# Reaching the Final Endgame for Constant Waves of COVID-19

**DOI:** 10.3390/v14122637

**Published:** 2022-11-25

**Authors:** Norman Arthur Ratcliffe, Helena Carla Castro, Marcelo Salabert Gonzalez, Cicero Brasileiro Mello, Paul Dyson

**Affiliations:** 1Biology Institute, Universidade Federal Fluminense, Niterói 24210-130, RJ, Brazil; 2Department of Biosciences, Swansea University, Singleton Park, Swansea SA2 8PP, UK; 3Institute of Life Science, Medical School, Swansea University, Singleton Park, Swansea SA2 8PP, UK

**Keywords:** SARS-CoV-2, Omicron, COVID-19 elimination, vaccines, nasal therapies, antiviral drugs, new therapeutic options, political and social hindrances

## Abstract

Despite intramuscular vaccines saving millions of lives, constant devastating waves of SARS-CoV-2 infections continue. The elimination of COVID-19 is challenging, but necessary in order to avoid millions more people who would suffer from long COVID if we fail. Our paper describes rapidly advancing and innovative therapeutic strategies for the early stage of infection with COVID-19 so that tolerating continuing cycles of infection should be unnecessary in the future. These therapies include new vaccines with broader specificities, nasal therapies and antiviral drugs some targeting COVID-19 at the first stage of infection and preventing the virus entering the body in the first place. Our article describes the advantages and disadvantages of each of these therapeutic options which in various combinations could eventually prevent renewed waves of infection. Finally, important consideration is given to political, social and economic barriers that since 2020 hindered vaccine application and are likely to interfere again with any COVID-19 endgame.

## 1. Introduction

In August 2022, waves of Omicron variants BA4 and BA5 around most of the World seem to be waning, although additional cycles of COVID-19 are being projected for the autumn/winter of 2022/2023 [1]. The uptake of vaccines has saved 10s of millions of lives but devastating cycles continue due to delays in developing alternative innovative preventative strategies [2]. In fact, the stunning success of vaccines has been equally matched by the rapid evolution of the virus with transmissibility increasing alarmingly for Omicron variants [3]. The majority of the vaccines have also targeted the Spike region of SARS-CoV-2 [4] in a form of monotherapy that inevitably selects for resistant virus variants to the vaccines. In addition, whether injected vaccines will ever end future recurrent cycles of infection is challenging. It is vital that optimal therapeutic practices are developed and adopted urgently to frustrate future devastating variants of SARS-CoV-2. Many previous reviews of aspects of COVID-19 have been published [5,6,7,8,9] but few have considered how the combination of potential therapeutic options could eventually prevent renewed waves of infection, as well as political, social and economic barriers likely to continue to hinder this COVID-19 endgame. This paper does not purport to be a comprehensive review of COVID-19 from 2019 until 2022 but concentrates on the early stage of invasion and the present- day therapeutic situation with possibilities in the future.

## 2. Intramuscular Vaccination

Intramuscular vaccines protect the lower respiratory system and lungs although, initially, transmission and infection by SARS-CoV-2 occurs mainly through the upper airways where Omicron variants have stronger infectivity [10]. Together with public health measures, injected vaccinations have been the main treatment option for preventing COVID-19. These intramuscular injections, however, largely induce serum immunoglobulin G1 (IgG1) protecting the lower respiratory tract rather than the strong antiviral IgA produced after nasal or mucosal vaccination [11]. Therapeutically, therefore, intranasal vaccines and drugs developed against early disease transmission and infection stages show more potential for interrupting COVID-19 cycles by blocking entry into the body [12,13,14].

Massive efforts were initially necessary to save lives by developing and successfully applying intramuscular vaccination but these distracted from the urgent subsequent need for alternative and combination therapies to end continuing disastrous cycles of COVID-19 infection, e.g., [15,16]. Significant delays of more than one year occurred in expanding research into new antiviral drugs and innovative methods for vaccine delivery [17,18,19]. Following several waves of COVID-19, however, the shortcomings of injected vaccines have finally been acknowledged due to the ease with which the virus can with mutation evade the immune response and this has prompted alternative strategies for COVID-19 therapy [2,3]. One such strategy is to modify present vaccines for treating new Omicron variants and use these as booster injections for previously primed individuals. Data indicate that this would give a wider breadth of antibodies against present and future variants, but would still defend against advanced disease and death [20]. Moderna and Pfizer recently had their bivalent vaccines approved for use as boosters in the UK and USA. These contain half from the original vaccine of 2020 and half targeting Omicron, and have been shown to generate good immunity towards both BA.4 and BA.5 [21]. Uncertainty, however, has existed as to whether we should continue to work behind viral evolution and chase our own tails or develop pan- coronavirus, broad-spectrum vaccines and anticipate future variants [21,22]. Both options, however, are being pursued with the bivalent vaccines now approved and development of pan- coronavirus vaccines with diverse targets and delivery strategies becoming a top research priority (Table 1) [23]. The Coalition for Epidemic Preparedness Innovations (CEPI) and the US National Institute of Allergy and Infectious Diseases (NIAID) have invested USD 243 million for developing pan-coronavirus vaccines [23]. There are many possible strategies for these vaccines including mRNA- protein nanoparticle-based technologies and mosaic approaches with multiple receptor-binding domains (RBDs) from humans and other animals. Some of these vaccines will only target the spike proteins, while others may interact with different components of the virus proteome. For pan-coronavirus vaccines to work, they probably must elicit both strong neutralizing and multiple T cell responses [23]. Finally, in these multiple antigen vaccines scientists have found that inclusion of components of the virus nucleocapsid elicit a good T cell response. The recent report of a universal influenza vaccine giving broad cross- protection against antigenically diverse influenza A and B viruses in young and aged mice is a reason for some optimism for the successful development of such a vaccine against multiple COVID-19 variants [24].

## 3. Intranasal Vaccination Therapy

One alternative strategy now being intensely researched is nasal vaccination (Table 2) [14]. Clinical trials are in process with nasal vaccination with the results eagerly awaited due to an urgent need to end recurrent waves of COVID by blocking breakthrough infections of Omicron, and future variants, at the portal of entry into the body [22]. There is, however, no guarantee for the clinical efficacy of intranasal vaccines against new Omicron variants since the majority of vaccines target the S2 region of the RBD spike which is subjected to multiple viral mutations [22]. Nasal vaccination may also be problematical as levels of neutralizing antibodies against pseudoviruses expressing eight variant and subvariant S proteins had the lowest values for Beta and Omicron SARS-CoV-2 [6]. Also of great relevance is the recent work with humans describing “hybrid-immune-damping” by which the previous vaccination and infection history have profound negative effects on subsequent immunity [25]. Many trials on nasal vaccination utilise small animal models. However, most short- lived, specific pathogen-free (SPF) laboratory animals do not have heterogenic immune histories, consequently, results of these trials may not truly reflect human responses [26]. This may explain the failure of a nasal vaccination clinical trial by Altimmune and the recent report of the failure of nasal vaccination in humans [27] The need for caution in interpreting data from SPF laboratory models is also illustrated by immunizing influenza into either SPF or “dirty” mice, with results showing that the SPF animals had enhanced immunity and protection to challenge compared with the dirty animals [26]. Since many influenza and COVID-19 vaccines are also initially evaluated in SPF mice, then use of such animals may limit the relevance of the results in translation to the human situation [26]. One recent needle-free, intranasal, multicomponent, mucosal bacteriophage (phage) T4-based COVID-19 vaccine gave robust cellular and humoral immune responses in mice against multiple variants [28], and ideally the results of this will transfer to humans too. Despite these concerns, hopefully, successful clinical trials of nasal vaccination will be reported soon. This could be a game changer allowing the transport and application of cheaper vaccines to poorer and more remote regions without the need of refrigeration or specialists for application. In addition, nasal vaccinations might be more acceptable to vaccine-hesitant people. In fact, during the writing of this review, Bharat Biotech International, India, has gained one of the first globally obtained approvals for an intranasal vaccine for its COVID-19 vaccine, iNCOVACC (BBV154). This approval was granted by the Central Drugs Standard Control Organisation (CDSCO) for Restricted Use in Emergency Situations for usage in people aged 18 years and older. Coincidentally, at about the same time, the Chinese company CanSino Biologics in Tianjin has been given approval for its inhaled COVID-19 vaccine for use as a booster dose in China. Both vaccines use a harmless adenovirus viral vector to transfer SARS-CoV-2 genetic material into their hosts. No phase III data from clinical trials has been published although it is claimed these studies have been concluded [29].

## 4. Non-Vaccine Nasal Therapy

In parallel with nasal vaccination, the approach to nasal therapy needs to be widened to consider alternatives to targeting the virus RBD spike with vaccines. Non-vaccine nasal therapy is an alternative innovative technique potentially providing protection following any compromise of nasal or intramuscular vaccinations. Such failure is likely to occur during waves of present or future Omicron variants. In non-vaccine nasal therapy, the host proteins rather than the viral proteins are targeted [30] and, unlike most present vaccines targeting the virus spike protein, the host proteins are unlikely to mutate extensively and mediate virus escape responses [31]. One example is a small molecular protease inhibitor (N-0385) blocking TMPRSS2 (involved in viral entry) in host cells and also subsequent infection, when applied intranasally in mice, by a broad spectrum of human coronaviruses, including Omicron variant B.1 and other respiratory viruses [31]. Additional examples are the use of ACE2 decoys or ACE2-Fcs (fragments crystallizable of antibodies) to bind SARS-CoV-2 spike proteins, inhibit virus uptake and lung damage as well as to enhance a more robust immune response [32,33]. These ACE2 decoys cross-neutralise COVID-19 variants including Omicron and can also be delivered intranasally. There are, however, suggestions that increased levels of circulating ACE2 molecules may affect cardiovascular disease [34]. Another possible host target includes furin, a type 1 membrane-bound protease with high expression in the lungs and associated with ACE2 activity. The spike protein of SARS-CoV-2 has a cleavable furin-like site involved in infection of the host cells. Several small molecular furin inhibitors have been shown to interfere with viral infection [35]. There are many other potential host protein targets with some clinical trials undertaken but this therapy needs much additional research and is challenging as not only are new host drugs needed (as suggested above) but the safety of delivery via the nasal route also has to be confirmed [13,35].

Alternative nasal therapy using simpler, much cheaper, available, non-vaccine molecules has also been described. For example, the nasal application of nitric oxide (NO) has been shown recently, in a randomized, double-blind, placebo-controlled phase III clinical trial with newly COVID-19-infected patients, to be safe with treated subjects negative for the virus by four days before the placebo controls [36]. In addition, NO treatment resulted in a 7⋅4 fold higher viral RNA reduction compared with the placebos at 48 h. NO-application is not expected to result in the development of drug resistance or affect other therapies. In fact, NO treatment has been recommended for use with antiviral drugs as a potent therapy for COVID-19 [36,37]. Despite nasal sprays for NO being rolled out internationally, additional clinical trials are still underway with NO used by inhalation or as sprays [38]. Cheap, safe, easily applied effective non-vaccine nasal sprays/inhalants for the COVID-19 virus should be available to poorer nations globally before too long. The eventual advantages of this non-vaccine nasal therapy approach, however, are that they avoid the potential side-effects of vaccines and have broad-spectrum activity. Such side effects are usually mild and include pain in the injected arm, tiredness, headaches, chills, nausea, upset stomach and flu-like symptoms which resolve after 1–2 days [39]. Like nasal vaccination, they also should be stable, not requiring specialists for application and be more acceptable to vaccine-hesitant people.

## 5. Antiviral Drugs

Most clinical trials on antiviral drugs for COVID-19 have been completed for hospitalized patients. One of the largest, the Recovery platform, identified three positive therapies, namely, dexamethasone, tocilizumab, and the casirivimab/imdevimab monoclonal antibodies, and excluded six other therapies including azithromycin, convalescent plasma, and hydroxychloroquine [13].

Clinical trials of drugs for early disease treatment, however, are more problematical as strategies have not been in place to optimise these studies. For example, it is necessary to identify patients most likely to deteriorate clinically for such trials [13]. The failure to rationalise these trials has also led to numerous studies with too few patients and with identical drugs repeatedly tested as in the case with over 150 trials reported of hydroxychloroquine [13]. In addition, new drugs were not developed as research efforts concentrated on testing repurposed drugs to save time and money.

For early treatment of non-hospitalized COVID-patients, the USA Food and Drug Administration (FDA) has only approved remdesivir, an RNA polymerase inhibitor, made by Gilead, although it has also given Emergency Use Authorizations (EUA) for Paxlovid (ritonavir-boosted nirmatrelvir), molnupiravir, and certain monoclonal antibodies (mAbs) [40]. Results of the use of remdesivir are contradictory but in severe COVID-19 cases early use is of greater benefit compared with later treatment [41], although reduced deaths in non-ventilated COVID-19 patients requiring oxygen supplementation have also been reported [42]. Paxlovid, developed by Pfizer, is taken orally, and is a protease inhibitor of the viral mPRO protease which is involved in viral replication. Such protease inhibitors will probably have less chances of viral evasion than highly specific monoclonal antibodies and therefore retain effectiveness against a range of emerging variants [43]. One study with nirmatrelvir (Paxlovid component) given to patients confirmed to have COVID-19 showed a 67% and 81% reduction in COVID-19 hospitalisations and mortality, respectively, in patients 65 years and over, while no such benefit occurred with younger 40–64 year olds [44]. Molnupiravir is a ribonucleoside analogue developed by Merck, taken orally, and inhibits SARS-CoV-2 replication. These two drugs should be taken within 5 days of symptoms and then only used for 5 days. Many questions remain regarding their use in combination therapy and possible interactions with other drugs being used by immunocompromised patients [45]. In addition, a monoclonal combination called Evushield from Astra Zeneca has also been granted EUA and is authorized as a pre-exposure medication for individuals with compromised immune systems or showing severe reactions to vaccines [40].

The good news is that there are now hundreds of drugs on trial against different stages of COVID-19 and many of these are in the NIH Active Programme and due to report this year [45,46].The focus in this review is on early stage antiviral drugs and these can be divided into those acting directly against the virus and those interacting with the host molecules [13]. Some of the latter were described above in Section 4 “Non-vaccine Nasal Therapy” but additional antiviral drugs, potentially applied orally or nasally, are under investigation. These include inhibitors of cathepsin L to reduce viral entry into cells and baricitinib an inhibitor of serine-threonine protein kinases involved in viral endocytosis [13]. Another, approach has been with repurposing selective serotonin reuptake inhibitor (SSRI) antidepressants such as fluoxetine and fluoxamine which not only have anti-proinflammatory cytokine properties but may also reduce symptoms in mild cases of COVID-19 [47]. Regarding drugs acting directly against the virus these include remdesivir and galidesivir blocking the viral RNA polymerase and inhibiting replication. The virus structural proteins are also potential drug targets especially the membrane (M)-proteins involved in virus assembly [13]. Reference to The National Institutes, COVID-19 “Treatment Guidelines of Health” [48] and The World Health Organization’s “Living Guideline” for COVID-19 drugs [49] describes many additional antiviral drugs reported for different stages of COVID-19 and their developmental status.

## 6. Discussion

### 6.1. Bringing It All Together to Block New Waves of COVID-19

Many people believe that elimination of COVID-19 is unlikely and we should learn to live with the virus. The virus, however, is still killing hundreds of people per day in the USA and due to illness keeping 500,000 people out of the work force [50,51]. Even people with mild or no symptoms can develop long COVID and serious health problems [52,53]. The information presented in this paper indicates that therapeutic scenarios are rapidly advancing and that tolerating continuing cycles of infection should not be acceptable. New vaccines with broader specificities are in the pipeline [21] as are innovative nasal therapies, e.g., [54] and antiviral drugs [45,46] which will prevent COVID-19 entering the body in the first place. The combination of these would form a formidable challenge to new COVID-19 variants. Unfortunately, these combinations, when available, would be very expensive and exacerbate treatment inequalities. Speculation of possibilities for preventing COVID-19 at the initial disease transmission and infection stages to end new cycles of virus are expanding and could include some of the following (with an indication of likely timelines):1.The injection of modified bi-valent (booster) or poly-valent (2–3 yr?) vaccines would prevent serious illness but may not prevent initial infection but could be given with boosters of nasal therapy/vaccination. Expensive, needing trained personnel.2.Nasal vaccination alone with a by- or poly-valent vaccine since nasal vaccination has been shown to elicit both nasal mucosal and systemic immune reactions. This would be relatively cheap and convenient for global use with outpatients (1–2 yr?).3.Non-vaccine nasal therapy against host proteins might be cheap and less likely to elicit virus breakthrough and with potential for use against a range of respiratory viruses and a global distribution for outpatients (3–5 yr?).4.Antiviral drugs, with many in the pipeline, could lead to new therapeutic strategies, yet to be thought of, with or without vaccines Current approved antiviral drugs for targeting early infection stages are expensive although cheaper and broad-spectrum drugs are in development (1–2 yr?) (e.g., [55]).5.Alternative nasal therapy using simpler, non-vaccine molecules, for example, the nasal application of nitric oxide (see Section 4. “Non-vaccine Nasal Therapy”, above) could be a significant development in terms of cost, convenience and inability of the virus to become resistant. NO could be used with other therapies such as antiviral drugs or vaccines (see Section 7 Conclusion and Future Prospects).

### 6.2. Political/Social Influences

The application of science alone is partially capable of eliminating COVID-19 although the success of vaccination against COVID-19 has been greatly influenced by both political and social factors. During the first year of COVID-19 vaccination it is estimated to have reduced total deaths globally by 63% (19·8 million of 31·4 million deaths) [56]. However, the same research also showed that because of low rates of vaccination (5.84%) in lower income countries (mainlyAfrica) compared with high income countries (68.8%), only 466,400 deaths were avoided [56]. These differences reflect inequalities of vaccine supplies due to political, social and economic determinants that have and continue to seriously hinder the successful development and application of COVID-19 therapies in poorer countries. These determinants have affected vaccination rates in most other countries too and have been far ranging to include inequalities in vaccine distribution, suspicion that vaccines contain tracking microchips, premature government easing of safety mandates, and the crass stupidity early on in the pandemic of some world leaders not wearing masks and thereby sending the wrong message about prevention. People have been the victims of misinformation and refuse vaccination and public health advice or suffer from COVID-19 fatigue. The end result is that huge numbers of people still have no access to vaccines, with ca. 30% of the global population, yet to receive a single vaccine dose [57]. Scientists advising governments and funding bodies, the media and so-called premier journals were also transfixed for 1 year on injected vaccines [12,17], despite widespread calls to broaden the development of antiviral strategies. This resulted in the paucity of options to treat the early stages of COVID-19 and spawned the massive waves of viral outbreaks generating new variants and the continued presence of populations susceptible to infection. A recent survey identified the poor uptake of antivirals and estimated that 100,000 to 150,000 lives could be saved next year (2023) in the USA alone by more widespread use of antivirals [58]. The estimate for cost in lives of the delay in developing antiviral drugs during the period 2020 to 2021 should be considered too.

The main challenge of antiviral drugs once developed is that they require application as soon as possible after acquiring the infection to act directly on viral replication, but delayed administration of drugs, for one reason or another, may result in a lack of effectiveness of such therapy [59,60].

## 7. Conclusions and Future Prospects

We should not accept the present high levels of COVID-19 and throw caution to the wind so that it is essential to continue to develop innovative therapies as quickly as possible. Deaths from COVID-19 up to 13 September 2022 were 4 to 10 times higher than influenza depending on the published source consulted, e.g., [61,62]. Even now, a new variant of Omicron, BA 4.6 is spreading in the UK, USA and other countries and has an R346T/S/I mutation giving heightened resistance to Evushield monoclonal antibodies used for immunocompromised people [63].

Therapies for early stages of COVID-19 are now developing rapidly with many clinical trials on modified vaccines, nasal vaccination and antiviral drugs due to report in 2022–2023 so the outlook for blocking new variants and repeated cycles is very promising. However, unlike US government’s Operation Warp Speed multibillion-dollar program, which rapidly developed the first COVID-19 vaccines, there are now problems with the necessary funding and commitment to advance some of these new therapies such as the pan-coronavirus vaccines [64].

New nasal therapies utilizing low cost, broad-spectrum molecules like NO are being developed and other antiviral nasal sprays have also been shown to have anti-COVID-19 activity in clinical trials, including the mimetic antimicrobial peptide, brilacidin, and astrodimer sodium [18,65,66]. Now is the time for conventional medicine and health watchdogs like WHO to realize these sprays have the potential for global use, including poorer counties, to reduce the burden of COVID-19 and thus the incidence of new devastating variants and waves of disease.

There are multiple political, social and economic barriers still hindering the end of new waves of COVID-19. Looking back at the previous record of governments in controlling COVID-19 gives an indication of the scale of these barriers that need to be addressed for the control of the present and future pandemics. A report by the Lancet describes some of the political and social shortcomings in managing the present COVID-19 pandemic [67]. The report describes “Widespread failures at multiple levels worldwide have led to millions of preventable deaths and a reversal in progress towards sustainable development for many countries” The failures occurred in prevention, transparency, rationality, standard public health practice, operational cooperation, and global solidarity. Most governments were too slow, ill-prepared, neglected the most vulnerable, and impeded by lack of trust and widespread misinformation. The report recommendations include:-1.Establishment of international vaccine-plus strategies to end the present COVID-19 pandemic and to ensure the safety of laboratory research into SARS-CoV-2 gene editing.2.Enhancement of multilateralism and the status of WHO as the lead global institution for emerging diseases. Ensure low/middle income countries have finance for research, development and production facilities. Strengthen health regulations and finance.3.Organize national pandemic preparedness plans.

In addition, the results of an UK enquiry into the pandemic are awaited, but in the meanwhile the British Medical Journal is also running a series which “examines how politicians used, and failed to use, evidence in response to the pandemic” [68]. There were many successes and failures but what all this soul-searching should do is help to rectify the many errors made so that COVID-19 waves of infection can finally be ended.

A recent statement on Tuesday September 6th from the White House COVID-19 co-ordinator Dr. Ashish Jha reflects the situation with the virus today:
“We now have all of the capability to prevent, I believe, essentially all of those deaths. If people stay up to date on their vaccines, if people get treated if they have a breakthrough infection, we can make deaths from this virus vanishing rare”.3860.

## Figures and Tables

**Table 1 viruses-14-02637-t001:** Selected pan-coronavirus vaccines in development.

Vaccine	Sponsor	Properties	Status
** *Variant-proof COVID-19 vaccines* **			
SpFN	US Army	Ferritin nanoparticle with prefusion-stabilized spike antigens from the Wuhan strain of SARS-CoV-2	Clinical
RBD–scNP	Duke University	Sortase A-conjugated ferritin nanoparticle with RBD antigens from early WA-1 strain of SARS-CoV-2	Preclinical
GRT-R910	Gritstone bio	Self-amplifying mRNA delivering spike and T cell epitopes	Clinical
hAd5-S+N	ImmunityBio	Spike and nucleocapsid antigens delivered via human adenovirus serotype 5 vector	Clinical
MigVax-101	MigVax	Oral subunit vaccine with RBD and nucleocapsid domains, adjuvanted	Preclinical
** *Pan-sarbecovirus vaccines* **			
GBP511	SK bioscience	Mosaic nanoparticle containing RBDs from SARS-CoV-1, SARS-CoV-2 and 1–2 bat coronaviruses	Preclinical
Mosaic-8b	Caltech	Mosaic nanoparticle containing RBDs from SARS-CoV-2 and 7 animal coronaviruses	Preclinical
VBI-2901	VBI Vaccines	Virus-like particles expressing prefusion spike of SARS-CoV-2, SARS-CoV-1 and MERS-CoV	Preclinical
** *Pan-betacoronavirus vaccines* **			
DIOS-CoVax	DIOSynVax	Needle-free injection of undisclosed antigens	Clinical
** *Other* **			
mRNA-1287	Moderna	mRNA encoding antigens from four human-infecting coronaviruses that cause common colds	Preclinical

From: Dolgin E. Pan-coronavirus vaccine pipeline takes form. Nat Rev Drug Discov. 2022 May; 21(5): 324–326. doi: 10.1038/d41573-022-00074-6 [23]. With permission of Nat Rev Drug Discov.

**Table 2 viruses-14-02637-t002:** Some of Intranasal COVID-19 vaccines in development *.

Developer (Location)	Vaccine Type	Delivery Method	Status
Bharat Biotech and (Hyderabad, India) and Precision Virologics (Washington University USA)	Viral vector; non-replicating	Intranasal (drops)	**Now approved in India.**
CanSino Biologics (Tianjin, China)	Viral vector; non-replicating (Aerosolized version of approved intramuscular vaccine)	Inhaled through nose and mouth	**Approved by Chinese regulators.**
Beijing Wantai Biological Pharmacy (Beijing)	Live attenuated	Intranasal (spray)	Phase III study under way in 40,000 people.
Razi Vaccine and Serum Research Institute (Karaj, Iran)	Protein subunit	Intranasal (spray)	Received emergency authorization in Iran in October 2021; in phase III trial (status unknown).
Codagenix (Farmingdale, New York) and Serum Institute of India (Pune)	Live attenuated	Intranasal (drops)	Phase II/III efficacy study in 20,000 people under way at undisclosed locations in Africa; part of the World Health Organization’s Solidarity Trial Vaccines.
Icahn School of Medicine at Mount Sinai (New York City) and Laboratorio Avi-Mex (Mexico City, Mexico)	Viral vector; non-replicating	Intranasal (drops or spray)	Phase II study under way in 396 people in Mexico City.
AstraZeneca (Cambridge, UK) and University of Oxford (Oxford, UK)	Viral vector; non-replicating (adenovirus)	Intranasal (spray)	Phase I study completed (both as first dose and as booster).
Meissa Vaccines (Redwood City, California)	Live recombinant	Intranasal (drops or spray)	Phase I study under way (both as first dose and as booster).
CyanVac (Athens, Georgia)	Viral vector; live, replicating	Intranasal (spray)	Phase I study under way.
Center for Genetic Engineering and Biotechnology (Havana, Cuba)	Protein subunit	Intranasal (spray)	Phase II study in up to 5000 participants in Cuba.

Source: Airfinity/*Nature* analysis. * Modified from: Waltz E. How nasal-spray vaccines could change the pandemic. Nature. 2022 Sep; 609 (7926): 240–242. doi: 10.1038/d41586-022-02824-3 [29].

## Data Availability

Not Applicable.

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
