# Peer review of "Reaching the Final Endgame for Constant Waves of COVID-19"

_viruses, 2022, doi:10.3390/v14122637_

Round 1

Reviewer 1 Report

This Review addresses a timely and important topic, namely recent developments in COVID-19 therapies. There are however, significant flaws with the manuscript in its current form.

General comments:

1) While the Authors' frustration with the situation is understandable, the manuscript is permeated with inappropriately emotional language, which one finds rather distracting. This starts right in the Abstract with "Many people believe that elimination of COVID-19 is unlikely and 10 we should learn to live with the virus, forgetting about the millions who would suffer from long COVID with this scenario." Extensive revision is therefore required to tone down the tenor of the manuscript to the appropriate standards of scientific - rather than creative - writing.

2) Some of the Authors' claims do not appear to have sufficient basis - i.e. they overstate the current evidence supporting clinical utility of NO.

3) The Authors fail to discuss some of the recent developments in drug repurposing for COVID-19, such as SSRI antidepressants.

Specific comments:

42 "." typo

52 "disastrous cycles of COVID- 52 19 infection" creative writing alert

53 "Significant delays of more than one year" - on the contrary, there is ample evidence that drug development has been considerably accelerated in the context of the COVID-19 pandemic

87 - "Table2" typo

99 - "However, most short- lived, specific pathogen-free (SPF) laboratory animals do not have heterogenic immune histories, consequently, results of these trials may not truly reflect human responses" it would be nice to have a link to some evidence for this

105, 109 - "Covid-19" or "COVID-19"? Needs to be consistent throughout the manuscript

130 - "our approach to nasal therapy" not clear why is it "their" approach?

157 - "rapidly eradicated 94% of SARS-CoV-2 in newly infected patients" there is no such data in references 28&29; reference 28 is a review/opinion piece, reference 29 merely provides some evidence that clearance of SARS-CoV-2 was accelerated

158 - "very significant" it can either be "significant" or "not significant"

254 - "Herein is a story of success and sadness" another outburst of creative writing

259 - "only 466, 400 deaths were avoided" is it 466 thousand or 466 and then 400?

262 - "These determinants have affected vaccination rates in 262 most other countries too and have been far ranging to include inequalities in vaccine dis- 263 tribution, suspicion that vaccines contain tracking microchips, premature government 264 easing of safety mandates, and the crass stupidity early on in the pandemic of some world 265 leaders not wearing masks and thereby sending the wrong message about prevention. 266 People have been the victims of misinformation and refuse vaccination and public health 267 advice or suffer from COVID-19 fatigue Before COVID-19, who would have imagined that 268 wearing a mask would be politicized ?!"

perhaps the Authors ought to log off Facebook for a while

299 - "New nasal therapies utilizing low cost, broad-spectrum molecules like NO could be 299 game changers." the Authors do not provide enough evidence to back such confident statements

Reviewer 2 Report

The present review is well designed and timely. However, the reviewer has a major concern that many such reviews related are available, and authors should appropriately cite the reviews and recent references, especially, focusing on intranasal drugs, such as usage of NO nasal sprays. Authors mention the reduction of viral load by 94%, which is true, but please mention that this %age was observed within 24hrs and, even the viral load was reduced to 98% within 48hrs, within the same study..

Can authors also mention some of the side effects from the current available vaccines? It would be worth to add a separate paragraph to discuss to make it a complete review.  

Reviewer 3 Report

Rathcliffe et all present a review describing (in their own words) ”... rapidly advancing and innovative therapeutic strategies for COVID-19 so that tolerating continuing cycles of infection should be unnecessary and unacceptable in the future. …”, and “potential therapeutic options” towards the COVID-19 endgame”.

The article features a very brief introduction, starting outright with waves of omicron BA4 and BA5 variants, followed by some very general remarks about vaccination and associated limitations. In fact, throughout the article there is a mixture of detail information followed by some very general (often obvious or omittable) comments.

The article then discusses the following sections: Intramuscular vaccination, Intranasal vaccination therapy, Non-vaccine nasal therapy, Antiviral Drugs, followed by a Discussion largely devoted to a mention of vaccination, nasal therapies, new antiviral drugs, and a Conclusion/Outlook mainly discussion political and socio-economic situations.

A total of 58 literature references is given.

Table 1 lists “Selected pan-coronavirus vaccines in development” This table is taken from another (referenced) publication.

Table 2 presents “Some Intranasal vaccines in development”, apparently also adapted from a previous publication. The table also includes non-SARS-CoV vaccines which would be of less relevance to COVID-19.

Given the ambitious claim to offer an overview concerning the “Endgame for constant waves of COVID-19”, the paper offers very little information and just some random sort of detail. After reading this article, one has learned about selected items of treatment, but no consistent or comprehensive overview is offered.

Overall, the article seems hastily assembled and written, lacking care in the collection and presentation of promised information. In the last section, font type and size vary randomly.

Examples:

Line 50-52:      “Massive efforts were initially necessary to save lives by developing and successfully applying intramuscular vaccination but these distracted from the urgent subsequent need for alternative and combination therapies to end continuing disastrous cycles of COVID-

19 infection.”
            What are such “alternative and combination therapies”? In what way was science “distracted” from the search for COVID-19 therapy. A PubMed search for “COVID-19 therapy” lists > 109,000 publications during 2020 – 2022, so there does nto appeare to be a lack of effort.

Line 60 onwards: what are “bivalent” vaccines? Mixtures of vaccines, or antibodies with two different epitopes?

Is the development of pan-virus vaccines the only therapeutic option available (with one example from a mouse study as proof of principle)?

Since all vaccines target a specific structure on the virus (whether mono- or multivalent targets are used), why should these pan-Virus vaccines not lead to buildup of resistance?

Line 105 – 113:

“One recent needle-free, intranasal, multicomponent, mucosal bacteriophage (phage) T4-based COVID-19 vaccine gave robust cellular and humoral immune responses in mice against multiple variants [20], and ideally the results of this will transfer to humans too. Despite these concerns, hopefully, successful clinical trials of nasal vaccination will be reported soon. This could be a game changer allowing the transport and application of cheaper vaccines to poorer and more remote regions without the need of refrigeration or specialists for application. In addition, nasal

vaccinations might be more acceptable to vaccine-hesitant people.”

-          Why should these results transfer to humans??

-          “Despite these concerns” – no concerns were listed, so what is the use of this phrase

-          “… hopefully, successful clinical trials will be reported soon…” is this the hallmark of a “gamechanger” ?? Why would these vaccines be cheaper? Why would no specialist be needed for the application?

-          Is there any study showing that nasal vaccination is really more acceptable to vaccination-hesitant people? Any study showing grater acceptance of nasal vaccination should be reported here.

The authors completely fail to consider recent reports that nasal vaccines do not show the expected efficiency (see T. Carvalho Nature Medicine News 03 November 2022

Intranasal COVID-19 vaccine fails to induce mucosal immunity. https://doi.org/10.1038/d41591-022-00106-z, and E. Waltz, Nature News Feature 06 September 2022 “How nasal-spray vaccines could change the pandemic. Vaccines inhaled through the mouth or nose might stop the coronavirus in its tracks, although there’s little evidence from human trials so far. Nature 609, 240-242 (2022) doi: https://doi.org/10.1038/d41586-022-02824-3).

The section on antiviral drugs is extremely short and incomplete. Numerous classes of drugs and drug candidates are being tested for their activity against numerous essential proteins of SARS-CoV-2. The few selected examples consider re-purposed antiviral drugs.

The discussion essentially repeats the previous content, and the conclusion/outlook section discusses in very few sentences some political and social problems in the distribution of antiviral treatment on a worldwide level. While important, these considerations are relevant once therapeutic options exist (as was seen in the distribution of vaccines), but not a factor in the development of new vaccines.

For an effective review leading towards new “options in the COVID-19 endgame”, one would need  

1)      Table about existing vaccines, approved, and in development/clinical testing and their target proteins.

2)      An overview of existing antiviral drugs and their targets. Numerous studies are published, and reviews summarizing these. At least, the reader should be directed to this information.

3)      As much focus is on nasal therapy, this – very interesting – aspect should be treated with more systematic care. What options exist, what successful therapies are in the market.

4)      Briefly mentioned throughout the article is formulation – can nanoparticle conjugates or other delivery agents improve efficacy of nasal (and other) therapies?

The authors set the scope of their own article as describing “ … the advantages and disadvantages of each of these therapeutic options which in various combinations could eventually prevent renewed waves of infection.”

In the present form, the article only mentions a selected few options, discussing them superficially, and does omit the greater part of information that available from published scientific literature. It does not give an overview of the field and the options pertinent to the endgame against COVID-19.

The topic of this article is exciting and of great relevance. In a review article, this deserves a more comprehensive and thorough treatment.

Suggestion: reject for publication in the present form.

Round 2

Reviewer 3 Report

In the revision of the manuscript, the authors have made very few changes to the text – largely of qualifying (not to say cosmetic) nature, not adding new information, but adding some important references.

As a referee, I was frequently accused of not having read the text, or not understood what was said. The review text intended to give some examples of my general assessment of the manuscript. In that it was selective.

The authors’ reply in one of these examples states:

“These points have been dealt with above and would be covered in many previous reviews some of which have been included now. However, even ref 9, by Sri Rekha M et al. Covid 19: A Comprehensive Review, 2022), does not attempt to list approved vaccines since this information if readily available in WHO and FDA websites and numerous other reviews.”

However, what I would expect of a review on therapeutic possibilities for the future is precisely up-to-date information and not the requirement to search WHO and FDA websites or numerous other reviews (why then another one?) for information.

It has been made clear, that the concept of the authors was another. So at this point, it is not so much a matter of article; content, but of scope. The authors claim:

“This paper does not purport to be a comprehensive review of COVID-19 from 2019 until 2022 but concentrates on the early stage of invasion and the present- day therapeutic situation with possibilities in the future.”

In answer to their claim, there is a brief and selective coverage of:

Intramuscular vaccines

Intranasal vaccination therapy

Non-vaccine nasal therapy

Antiviral drugs

Political and socioeconomic considerations

These points are all valid and interesting but the compilation of information is not comprehensive enough to cover the “present-day therapeutic situation and possibilities in the future” adequately. Scope and depth of the review should be more. As an Opinion article, or Brief Update, this would qualify. As a review article, it is not helpful since it offers just the authors' assessment of some aspects of COVID-19 therapy.

Suggestion is "Reconsider after revision", asking for more comprehensive treatment of the topics.